# Overlapping qubits from non-isometric maps and de Sitter tensor networks

ChunJun Cao[1,2,3], Wissam Chemissany[4], Alexander Jahn [2,5] ✉ & Zoltán Zimborás [6,7,8]

The emergence of a local effective theory from a more fundamental theory of quantum gravity with seemingly fewer degrees of freedom is a major puzzle of theoretical physics. A recent approach to this problem is to consider general features of the Hilbert space maps relating these theories. In this work, we construct approximately local observables, or overlapping qubits, from such non-isometric maps. We show that local processes in effective theories can be spoofed with a quantum system with fewer degrees of freedom, with deviations from actual locality identifiable as features of quantum gravity. For a concrete example, we construct two tensor network models of de Sitter spacetime, demonstrating how exponential expansion and local physics can be spoofed for a long period before breaking down. Our results highlight the connection between overlapping qubits, Hilbert space dimension verification, degree-of-freedom counting in black holes, holography, and approximate locality in quantum gravity.

Although we often take for granted that the Hilbert space of $N$ qubits should be $2^N$-dimensional, the qubits we encounter in experiments are mere approximations of the ideal. In practice, the actual Hilbert space dimension of such physical qubits can be difficult to verify with limited computational resources and finite experimental precision. However, this verification problem is not to be lightly dismissed as a nuisance as it carries profound physical consequences even in theory. For example, in ref. 1 it was shown that if we only require certain quantum measurement outcomes to be reproduced approximately, it is possible to spoof even $N = O(\exp(\epsilon^2 n))$ qubits using only $n$ exact qubits as long as the $N$ qubits can overlap; that is, elements of the Pauli algebra of any two such qubits $P_i$, $P_j$ satisfy $\|[P_i, P_j]\| < \epsilon$. This spoofing is not without its limitations; for example, ref. 1 devised a verification protocol with poly($N$) complexity. Intriguingly, quantum gravity is widely understood to exhibit a qualitatively similar degree of freedom counting problem[2–6]. While the number of degrees of freedom in a local quantum field theory grows volumetrically, in quantum gravity we expect entropy to grow with surface area as exemplified by black

hole entropy[7,8] and holography[5,6,9]. This problem not only applies to the interior of black holes[10] but also to the vacuum when the spatial curvature is non-negative. Even in the context of AdS/CFT, the same is expected for a theory with sub-AdS locality. Although it is widely accepted in holography that the Hilbert space of quantum gravity should indeed be smaller than expected from the effective field theory (EFT), an explicit resolution of this degree of freedom counting problem remains open. In particular, this would require a mechanism that on the one hand reproduces the notion of locality in the EFT and on the other accommodates these degrees of freedom into a much smaller quantum gravity Hilbert space.

In this work, we stress the importance of approximate orthogonality in the context of quantum gravity and cosmology, e.g.,[10–14], and revisit this idea from a different perspective. We first draw upon an analogy between the Hilbert space dimension verification problem in quantum information and the degree of freedom counting problem in quantum gravity. Furthermore, we propose that the illusion of a bigger Hilbert space in local EFT can be reconciled with

[1]Joint Center for Quantum Information and Computer Science, University of Maryland, College Park, MD, USA. [2]Institute for Quantum Information and Matter, California Institute of Technology, Pasadena, CA, USA. [3]Department of Physics, Virginia Tech, Blacksburg, VA, USA. [4]David Rittenhouse Laboratory, University of Pennsylvania, Philadelphia, PA, USA. [5]Department of Physics, Freie Universität Berlin, Berlin, Germany. [6]QTF Centre of Excellence, Department of Physics, University of Helsinki, Helsinki, Finland. [7]Algorithmiq Ltd, Helsinki, Finland. [8]HUN-REN Wigner Research Centre for Physics, Budapest, Hungary. ✉e-mail: a.jahn@fu-berlin.de

the smaller Hilbert space of quantum gravity if the "qubits" that make up the EFT are approximate qubits that overlap. In the field-theoretic language, it means that for spacelike separated observables $O(x), O(y), [O(x), O(y)]|\psi\rangle \neq 0$, such that microcausality can be broken for some states $|\psi\rangle$, e.g., those far away from the vacuum. Though such non-locality may seem unphysical, it is well-established that the algebra of local observables for a spacetime region is no longer well-defined in the presence of gravity[14–18], as we visualize in Fig. 1. Furthermore, a "verification protocol" of the quantum degrees of freedom promised by a local EFT, such as creating massive states with low energy density, fails due to black hole formation. To further sharpen this proposal, we introduce a framework for understanding approximate, overlapping qubits which form an approximate algebra of observables for a local region in spacetime using general non-isometric maps.

## Results

### Overview and previous constructions

Our construction of overlapping qubits follows a similar spirit as[1,10], but combines both perspectives and builds upon their merits. We put forward a generalization of the mode transformation used in the original overlapping qubit construction[1] to express non-isometricity in the language of commutators. On the one hand, this introduces more versatile and intuitive methods to overlap qubits; on the other hand, it endows richer local structures absent in ref. 10. We show by example that by endowing more structures to a non-isometric map than their Haar-random counterparts, more physically relevant scenarios can be constructed where overlap can depend on distance and that the familiar physics such as locality, low energy dynamics, and unitarity can be approximately preserved. In addition, previously state-independent quantities such as microcausality, enforced by vanishing commutators for spacelike separated observables, become state-dependent. We claim that the overlap between such approximate qubits in some constructions may in part be identified with the weak non-locality expected in gauge theory, such as the non-local effects introduced by gravitational dressing. We further identify the discernible departure of the spoofed dynamics from the local EFT with the effects of gravity. As a consequence, the state-dependent commutation relations in the spoofed system are analogous to the effects

from gravitational backreactions that shift the underlying spacetime geometry. When the spoofing breaks down and the state-dependent contribution term reaches $O(1)$, this corresponds to the breakdown of the local EFT. This is analogous to strong gravitational effects dominating and the background causal structure being significantly altered, e.g., by the formation of black holes when accessing the massive states in the EFT.

Beyond the above statements which hold in generality, we also apply the formalism to a tensor network toy model of de Sitter (dS) spacetime[19]. This spacetime, in addition to being physically relevant for our own expanding universe, presents a case where the tension between EFT and quantum gravity degrees of freedom counting is at its strongest. Tensor networks with suitable geometries are a natural discretization of the maps between EFT Hilbert spaces on different time-slices, though it is at present unclear if they can also describe features of full, nonperturbative quantum gravity. For a system with dS entropy $S_{dS}$, we show that it is possible to construct a tensor network model with a strongly complementarian picture following Susskind[20,21] where a seemingly global dS space can arise from a Hilbert space dimension of only $O(\exp(S_{dS}))$ where each static patch has entropy $O(S_{dS})$ in the effective description. We argue that for any local observer living in the dS universe, the EFT description of global dS may be approximately preserved for up to time $T \sim S_{dS}$ in units of Hubble time. This can be an exceedingly long time for a dS universe with our current cosmological constant, where $S_{dS} \approx 10^{122}$. However, such an effective description is necessarily unstable beyond time $T$, where a transition to a new effective background geometry for the EFT must occur, consistent with the expectation of ref. 22. Complementary to this line of thought, it has been proposed that the Hilbert space of quantum gravity states that remain asymptotically dS is finite-dimensional[23]. We also construct an alternative, weakly complementarian picture, where the number of degrees of freedom would grow linearly instead of exponentially with time and where the notion of local physics does not break down at late times. Compared to past constructions[1,13] based on random maps, the tensor network examples also demonstrate a connection between internal structures of the non-isometric maps and the emergence of spatial locality among overlapping qubits. We then discuss a few observations on complexity, tensor networks, and von Neumann algebras.

Our contributions are organized as follows. We provide a construction of approximate overlapping qubits based on non-isometric maps, which is different from both refs. 1 and 10,13. We then discuss the consequences of spoofing real qubits with overlapping ones. We further describe how an expanding spacetime can be spoofed by overlapping qubits using two different MERA tensor network toy models. In the first model, the fundamental Hilbert space is finite, though the volume of time slices in the EFT description (and hence, the number of overlapping qubits) grows exponentially in time. This effective description is then only valid for finite time and the overlap is distance-dependent. A similar distance dependence can be observed in the second dS model, which only describes the EFT in the local patch, leading to a scenario in which spoofing can persist indefinitely. Further details of the proof and discussions are given in the Supplemental Information.

### Physics of approximate overlapping qubits

Let $n$ exact fundamental qubits—or more generally, qudits—live in a Hilbert space $\mathcal{H}_n$. It is possible to fit $N > n$ approximate *overlapping qubits* into this Hilbert space, over which a quasi-local effective theory that is approximately unitary can be defined. In what follows, we generalize the explicit proposal for overlapping qubits of ref. 1 twofold: First, we do not require each individual qubit algebra to be exact, but allow small overlaps between the local (approximate) Pauli operators as well. Second, rather than merely considering mode transformations on $\mathcal{H}_n$ to construct the overlapping qubit algebra, we

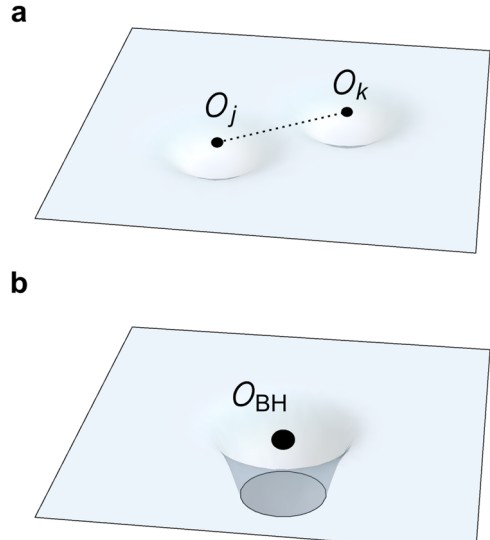

**Fig. 1 | Modifications to the Hilbert space of effective field theory (EFT) by quantum gravity. a** Two EFT operators $\mathcal{O}_j$ and $\mathcal{O}_k$ will, as a result of gravity, lose exact commutativity, introducing small non-locality (dashed line). **b** Any operator $\mathcal{O}_{\mathrm{BH}}$ producing a black hole state restricts the semi-classical EFT background, leading to a Hilbert space truncation.

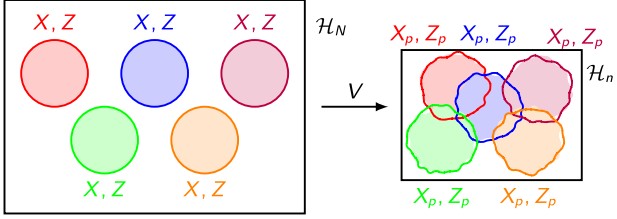

**Fig. 2 | Hilbert space compression.** The non-isometric map $V$ maps nominally exact qubits (circles) in $\mathcal{H}_N$ onto approximate overlapping qubits (jagged circles) in a lower-dim. $\mathcal{H}_n$ where $\{X_p, Z_p\} \approx 0$.

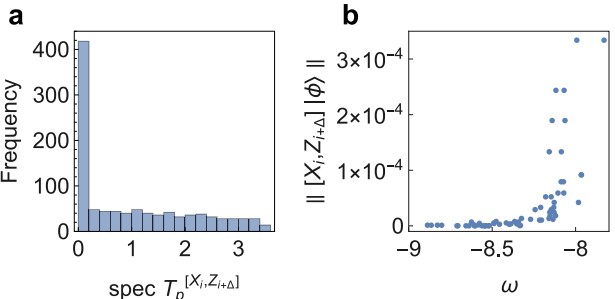

**Fig. 3 | State-dependent operator overlap. a** For $V$ being a low energy truncation of a 1d critical Ising CFT, we show the eigenvalue distribution of $T_p^{[X_i, Z_{i+\Delta}]}$ as a histogram, where $X_i$, $Z_{i+\Delta}$ are local Pauli operators separated by a fixed spatial distance $\Delta$. **b** shows a correlation between the amount of microcausality violation and the "energy" $\omega$ of each state $|\phi_k\rangle$ labeled by different $k$s, such that more energetic states generally incur greater violations, consistent with our intuition. The energy $\omega$ is defined by a truncated critical Ising Hamiltonian. See Supplementary Note 3 for detailed definitions.

consider a more general surjective linear map $V : \mathcal{H}_N \to \mathcal{H}_n$. This map, whose action is visualized in Fig. 2, is necessarily non-isometric for $N > n$. Let $Q \in L(\mathcal{H}_N)$ and $Q_p = VQV^\dagger \in L(\mathcal{H}_n)$ be bounded linear operators over their respective Hilbert spaces and $|\psi_p\rangle = V|\psi\rangle$. We say that $Q_p$ acts on an approximate overlapping qubit if $Q$ is supported on a qubit subsystem in $\mathcal{H}_N$. The action of $\prod_i Q^{(i)}$ in the effective theory can be *spoofed* with respect to $|\psi\rangle$ if $\langle\psi_p|\prod_i^M Q_p^{(i)}|\psi_p\rangle \approx \langle\psi|\prod_i^M Q^{(i)}|\psi\rangle$. It is easy to see that for suitable maps $V$, operators, and states $|\psi\rangle$ (potentially restricted to a physical subspace), unitarity and algebraic structure are also approximately preserved, with $\langle\psi|U^\dagger U|\psi\rangle \approx \langle\psi_p|U_p^\dagger U_p|\psi_p\rangle$ and $\| [Q, Q']|\psi\rangle \| \approx \| [Q_p, Q_p']|\psi_p\rangle \|$, and similarly for the anti-commutation relations. More generally, one can identify such state-operator pairs given a form of $V$.

**Remark 1.** (*Approximate homomorphism property*): Let $P = V^\dagger V$ be a truncation map. The set of processes $\mathcal{S} = \{\prod_i^{\le M} Q^{(i)}\}$ is well-spoofed, i.e.,

$$\left\langle\psi_p\right| \prod_i^{\le M} Q_p^{(i)} \left|\psi_p\right\rangle \approx \left\langle\psi\right|\prod_i^{\le M} Q^{(i)}\left|\psi\right\rangle, \qquad (1)$$

if $\forall S \in \mathcal{S}$, $\| PS|\psi\rangle - S|\psi\rangle \| \approx 0$.

In other words, a sufficient condition for identifying well-spoofed processes is if the string of operators $Q^{(i)}$ do not take $|\psi\rangle$ out of the subspace that is approximately preserved by $V$. In general, given a $V$ (e.g., acting on the Hilbert space of black holes[10]) one can also identify other well-spoofed processes by solving a set of constraint satisfaction problems (see Supplementary Note 1 for details). For any such $V$, one can perform a singular value decomposition and write $P = \sum_k \lambda_k |\psi_k\rangle\langle\psi_k|$ with orthonormal set of states $\{|\psi_i\rangle \in \mathcal{H}_N\}$ (note that $P$ is not a projection in general). The processes preserved by spoofing naturally depends on the form of $V$. For example, suppose $V$ is a global truncation onto the low energy subspace of some local Hamiltonian, e.g., that of a local QFT, such that for some global energy cut off $\Lambda$, $\lambda_{k \le \Lambda} = 1$, then the processes $\prod_i Q^{(i)}$ that can be effectively spoofed are "low energy processes" which take $|\psi\rangle$ to states that have negligible support over the high energy subspace with $k > \Lambda$[24,25]. Such is expected for a local theory and associated operators as a consequence of UV-IR decoupling. A similar conclusion holds for our later MERA tensor network construction, as it prepares states with power-law correlations related to conformal field theories. If, on the other hand, $V$ is proportional to a Haar-random projection as in refs. 10,13, then with high probability, these subspace preserving processes are operators with subexponential complexity if $|\psi\rangle \notin \ker(V)$. Returning to general $V$s, we note that these approximate qubit operators also overlap. This overlap quantifies how well locality is preserved by the spoofing. For any $Q, Q'$,

$$[Q_p, Q_p']|\psi_p\rangle = V[Q, Q']V^\dagger|\psi_p\rangle + T_p^{[Q, Q']}|\psi_p\rangle, \qquad (2)$$

where $T^{[Q, Q']} = QTQ' - Q'TQ$ and $T = V^\dagger V - I$. If $[Q, Q'] = 0$, then the projected operators indeed overlap by an amount given by $\| T_p^{[Q, Q']}|\psi_p\rangle \|$. Depending on the spectral property $\mathrm{spec}(T_p^{[Q, Q']})$, we

can immediately deduce: (1) If $\exists\lambda \in \mathrm{spec}(T_p^{[Q, Q']})$ such that $\lambda < \epsilon$, then there exist states $|\psi_p\rangle$ for which the locality is approximately preserved, i.e., $\| [Q_p, Q_p']|\psi_p\rangle \| = O(\epsilon)$. For instance, this is the case if $Q, Q'$ acting on $|\psi\rangle$ are well-spoofed processes we examined above. (2) The value of the commutation relation is state-dependent if $\mathrm{spec}(T_p^{[Q, Q']})$ is non-flat. We find that both are the case for the $V$s we consider in this work, where the spectrum is non-flat with many of its eigenvalues are concentrated near zero (Fig. 3a). For example, we show that if $V$ is a low energy truncation of a 1d Ising CFT, then commutators of local observables with respect to low-energy states are relatively well-preserved. Interestingly, the states that give rise to larger microcausality violation also correlate with higher energies (Fig. 3b). See Supplementary Note 2 for details.

These observations carry physical significance in the context of quantum gravity. If $Q, Q'$ are gauge-invariant operators[18] that are spacelike separated, then the small but non-vanishing commutator of $[Q_p, Q_p']|\psi_p\rangle$ is consistent with them being weakly non-local by turning on gravity. Formally, $T_p^{[Q, Q']}$ plays the role of the gravitational correction, where it is generally a non-local operator straddling the local operators $Q, Q'$. Note that the gravitational correction can admit both perturbative and non-perturbative contributions. However, the perturbative effect may depend on the choice of the gravitational dressing[18]. On the other hand, changes to the state $|\psi\rangle$ would generally signal a different configuration for the stress-energy, which through gravity, leads to a different background spacetime geometry. In this sense, we should also expect the causal structure defined by the commutators to be state-dependent. Hence, we identify the departure of the actual physics obtained via spoofing from the local EFT on a fixed background with the effect of gravity or potentially new physics beyond local EFT-based predictions in the Standard Model or $\Lambda$CDM.

Finally, as the fundamental processes cannot spoof everything promised by the EFT in a much larger Hilbert space, there are regimes for which $\lambda$ is large and $T_p^{[Q, Q']}|\psi_p\rangle$ dominates. Such is also the premise behind the verification protocol of ref. 1. Then the effective description with $[Q, Q']|\psi\rangle = 0$ is not even approximately true. This is similar to the scenario where the effect of quantum gravity dominates and the local EFT is expected to break down. From Remark 1, we see that this may be because the effective processes now have large support over the subspaces that are being cut off. In the global energy truncation, it corresponds to considering $|\psi\rangle \in \ker(V)$, which is also known as the null space of the non-isometric map. This may happen in quantum gravity, for instance, when a large black hole has evolved long past the Page time. We show that this breakdown also occurs for EFT states at late times for the dS MERA model we introduce below.

## Overlapping qubits in de Sitter

As we have seen, overlapping qubits resulting from a non-isometric truncation of EFT Hilbert spaces produce effects that are in principle compatible with those of quantum gravity effects. However, we did not yet specify the form of such a non-isometric map or the physical setting in which it may occur. As a specific example to explore our idea, we now consider the case of dS spacetime, a model of an expanding universe. A long mystery of semiclassical dS spacetime has been the relationship between the EFT Hilbert spaces living on different time-slices: It appears that as the spatial volume increases exponentially with time $t$, so does the size of EFT Hilbert space. It has therefore been suggested that the usual unitary evolution rules of quantum mechanics have to be replaced by *isometric* evolution in the dS case[26,27]. In fact, such an isometric map appears to become neccessary in any dynamical spacetime with a UV cutoff[28]. A potential resolution to this problem is that while the EFT Hilbert space may be growing, the *fundamental* Hilbert space may remain constant, an idea similar to which has recently been applied to black hole evaporation[10]. Given the isometric evolution map $V$ of the dS setting, a non-isometric map to a smaller Hilbert space is readily available: It is given by $V^\dagger$, inducing a truncation $VV^\dagger$ on the EFT Hilbert space. This leaves the choice of an initial time-slice with which the fundamental Hilbert space is associated. Globally, there is only one coordinate-independent choice to make, the time-slice of minimal volume at $t = 0$ in the standard coordinates, as we show in Fig. 4. Note that in other spacetimes with time-dependent expansion or contraction, there may be several extremal-volume Cauchy slices, complicating the choice of the "fundamental" slice. As we show below, one can extend this proposal to time-slice subregions as well, where again a fundamental Hilbert space may be associated with a minimal surface.

To explore the concrete behavior of the non-isometric maps in dS in a finite setting, we will be considering a discretization of dS space-time in terms of the MERA tensor network, which possesses the same causal structure. We begin by reviewing the properties of this tensor network model.

## Review of de Sitter MERA

The *multi-scale entanglement renormalization ansatz* (MERA) is a tensor network ansatz designed to produce (ground) states of critical

Hamiltonians[29], used as lattice models of conformal field theories[30]. It is most commonly used for $1 + 1$-dimensional theories, where the tensor network takes the form of a planar branching tree with interconnections on each layer. Even though this geometry breaks translation invariance, the MERA can be used to efficiently simulate translation-invariant models[31]. Along the branching direction of the network a flow of *entanglement renormalization* can be identified[30,32]. As this resembles the radial direction of an AdS time-slice in AdS/CFT, it was proposed that the MERA is a discrete model of AdS holography[33]. However, it was later argued that the tensor network geometry of the MERA does not directly correspond to the spatial geometry of an AdS time-slice[34,35]. Rather than a spatial coordinate in a negatively curved space-time, another interpretation of this branching direction is that of a time coordinate in positively curved dS space-time[19,35–38]. In this picture, each MERA layer corresponds to a time-slice of an expanding universe, with each tensor representing a piece of spacetime with spatial and temporal extent of the order of the Hubble radius and Hubble (doubling) time, respectively. The MERA thus parametrizes the isometric map that appears between effective degrees of freedom on these time-slices[26–28] while incorporating causal constraints: The tensor network structure of the MERA, using the standard MERA constraints on unitary disentanglers and isometric branching tensors, reproduces the causal structure of the $1 + 1d$ global dS spacetime. In particular, for a $k$-branching MERA there exists a tensor network notion of a *static patch*, the part of the universe visible to a stationary observer: It is given by a wedge comprised of $k$ isometries and $k − 1$ disentanglers at each timeslice, as shown in Fig. 5 for $k = 2$. As in continuum dS, one can divide the entire space at $t = 0$ into two halves $A$ and $B$, each centered around a "pode" and an "antipode", i.e., two stationary observers. Keeping the causally accessible region for each observer by tracing along the timelike paths in the network yields two regions whose (proper) volumes are fixed under exponential expansion with $t$; the resulting spacetime volumes (red and blue regions in Fig. 5) are the two static patches of the initial observers. This is the discrete analog of the static patch in the tensor network model where it corresponds to a vertical strip of tensors, alternating between a disentangler and two isometry tensors.

In a scale-invariant MERA, the state inside the static patch also tends to a fixed point, yielding an expected outcome from the "cosmic no-hair theorem"[19]. For each spatial slice, the dS entropy in this model is upper bounded by the edge cuts that separate the interior of the patch from the exterior. As each tensor is of Hubble scale, we have the bond dimension $\chi \sim O(\exp(e^{S_{dS}}))$ for the MERA tensor network. Therefore, the entropy of each static patch is precisely $O(S_{dS})$, set by the bond dimension of the tensor contractions.

With the correct choice of tensors, the MERA also produces the ground state of many critical lattice models with a CFT continuum limit at $t → \infty$[30,31]. In the dS picture, the sites associated with that state are located at future infinity $\mathcal{I}^+$. As the symmetry group $SO(d, 1)$ is both obeyed by empty dS spacetime as well as a Euclidean CFT, it is tempting to identify this asymptotic CFT as a "boundary dual" of dS gravity. However, unlike AdS/CFT, the boundary lacks a time-like coordinate, precluding a dynamical dictionary, and the "duality map" between dS space-time and $\mathcal{I}^+$ is simply the causal evolution of the universe. It is worth noting that other interpretations of the MERA also exist. For example, when relating MERA to path integral geometries, it was argued that MERA network describes a light-like hypersurface in AdS spacetime[39]. As we make no connection with path integral geometries or the AdS/CFT correspondence, these interpretations are irrelevant for our purposes, where the salient feature of the MERA is its causal structure which matches discretized dS spacetime. We also point out that the critical state a MERA can be tuned to produce at $\mathcal{I}^+$ has no special meaning here and should not be confused with the usual emergence of a semiclassical bulk in AdS/CFT.

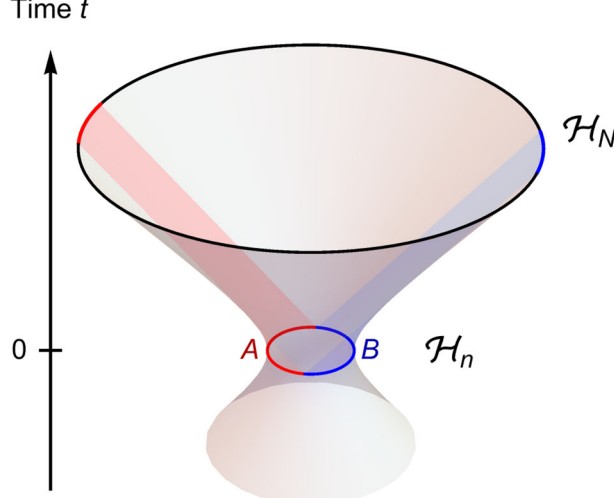

**Fig. 4 | Global De Sitter spacetime as a hyperboloid in a higher-dimensional Minkowski embedding.** The volume of time-slices increases exponentially with global time $t$, with a minimum effective Hilbert space $\mathcal{H}_n$ at $t = 0$. We propose a projection of effective Hilbert spaces $\mathcal{H}_N$ at $t > 0$ into the "fundamental" Hilbert space $\mathcal{H}_n$. Two causal patches with constant volume are shaded in red and blue, emanating from two complementary regions $A$ and $B$ at $t = 0$.

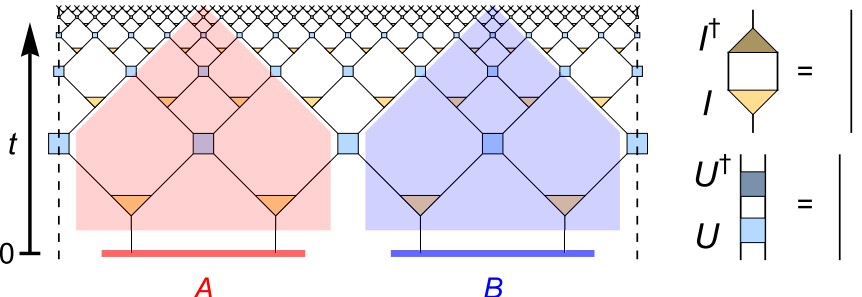

**Fig. 5 | De Sitter as a tensor network.** The causal structure of two-dimensional de Sitter spacetime $dS_2$ can be discretized with the MERA tensor network, consisting of "disentangling" unitaries $U$ (square) and isometric tensors $I$ (triangles) whose properties are shown on the right. The left and right edges of the the tensor network (dashed lines) are periodically identified. The "entanglement renormalization" direction of the MERA is identified with global dS time $t$. The two complementary static patches from Fig. 4 are shown here as well, again shaded in red and blue.

Using the MERA as a coarse-graining description of dS spacetime, one immediately realizes the proposed isometric time-evolution from a time-slice at earlier time to one at later time[27], in that the state produced by the MERA lives in a nominal Hilbert space whose size doubles at every layer. However, using suitable cuts through the MERA we can also interpret it as a map from this nominal, rapidly expanding Hilbert space to a much smaller "fundamental" Hilbert space. In this concrete tensor network setting, we can explore how such maps constrain the space of operators or states in the nominal Hilbert space. We now propose two such MERA-based map: A "global" proposal in which the fundamental degrees of freedom stay constant with time $t$, and a "local" proposal where their growth is merely linear, rather than exponential. We qualify that the dS/MERA construction only serves as a coarse-grained mapping for how the quasi-local EFT degrees of freedom on the (super) Hubble scale that increase in time emerge from a smaller "quantum gravity" Hilbert space where the fundamental theory can be highly non-local. It does not constitute a theory of dS quantum gravity.

Although we mostly make general statements about the MERA, it is also helpful at times to consider specific examples where disentanglers and isometries are fixed to particular values. For benchmarking purposes, we consider three explicit instances. For the first instance, we instantiate tensors of bond dimension $\chi = 2$ that optimize the ground state of the critical transverse field Ising model, whose continuum limit is the Ising CFT. The details of this construction are explained in Supplementary Note 4. We also consider a model with Haar-random unitaries selected as disentanglers and isometries with different choices of $\chi$. These examples are studied numerically. Finally, we consider a more general construction at large $\chi$ based on analytical arguments. Generally, we expect the large $\chi$ limit to be necessary for realistic models of dS spacetime, so that each static patch is described by sufficiently many parameters to resolve details below the Hubble scale.

## Global dS MERA

Global dS evolution, with the apparent Hilbert space degrees of freedom doubling every Hubble time, can be naturally discretized in terms of the MERA[19]. From the finiteness of dS entropy $S_{dS}$, complementarity, and dS holography, it has been suggested that the dimension of the quantum gravity Hilbert space for dS spacetime is finite[40–44] and given by $O(\exp(S_{dS}))$, which is proportional to the dS horizon area or the area of the spacelike extremal surface at $t = 0$. References[26,27] proposed that this Hilbert space is isomorphic to a subspace of states that do not lead to big crunch cosmologies. On the surface, this finite-dimensional description seems to be at odds with the semiclassical picture that has exponential growth in the spacetime volume. It is also unclear how the expanding spacetime should emerge from a finite dimensional Hilbert space. Here we suggest a possible construction by describing the

expanding global dS space with an increase of overlapping qubits, whose fundamental Hilbert space has dimension $O(\exp(S_{dS}))$. As a consequence, the apparent exponential expansion can last for a long time before the notion of locality breaks down. Such an entropic restriction is also consistent with previous studies of dS quantum gravity, the co-kernel of whose time evolution is restricted to states at the throat region at $t = 0$[38,40]. For a single static patch, this leads to the expected entropy $S_{dS}$.

To identify the overlapping qubits, we build a non-isometric map using the MERA. Let $\mathcal{H}_t \cong \mathcal{H}_{N_t}$ be the apparent Hilbert space associated with the global timeslice $\Sigma_t$ at time $t$ and $\mathcal{H}_{t=0} \cong \mathcal{H}_n$ be the fundamental degrees of freedom tied to the dS quantum gravity with $N_t \geq n$, where $\dim \mathcal{H}_{t=0} \sim \exp(S_{dS})$. We posit that dS quantum gravity is governed by some dynamical processes over $\mathcal{H}_n$, such that the effective description up to some time $T$ can be approximated by a sequence of non-isometric maps $V_t : \mathcal{H}_{N_t} \to \mathcal{H}_n$, each of which is given by a MERA cut off at time $t$ (Fig. 6a). Using $V_t$, we can identify the algebra of observables (e.g., the ones supported on each site) in $\mathcal{H}_t$ with an approximate algebra in $L(\mathcal{H}_n)$ using our overlapping qubit construction. This effectively compresses $O(\exp(t))$ number of qubits into an only constant number of qubits at the expense of introducing overlaps. Here let us focus on MERAs that are scale- and translationally invariant, i.e., the disentanglers (and isometries) at each layer are identical. By fixing the individual tensors, an explicit mapping $V_t$ can be constructed. As an isometric map, $V_t^\dagger$ identifies a subspace $\mathcal{C} \subset \mathcal{H}_t$. We can conclude from Remark 1 that operations preserving the subspace $\mathcal{C} \cong \mathcal{H}_{t=0}$ are well-approximated, thus recovering properties like the cosmic no-hair theorem in ref. 19 or those supported in ref. 27. Here $\mathcal{C}$ is a space of states with power-law correlations which one can obtain by only altering the initial conditions at $t = 0$. Different from random maps used in refs. 10,13 and the construction by ref. 1 based on the Johnson-Lindenstrauss (JL) lemma, the locality of MERA overlaps the qubits by an amount that depends on the proper distance.

**Proposition 1.** Let $O^{(i)}, O^{(j)}$ be local operators supported on qubits $i$ and $j$ respectively on $\Sigma_t$ separated by proper distance $|i - j|$ in units of Hubble radius in the MERA, then non-isometric maps can be constructed to produce overlap such that for some state $|\psi_p\rangle$

$$\left\| \left[ O_p^{(i)}, O_p^{(j)} \right] |\psi_p\rangle \right\| \lesssim \frac{\epsilon(t)}{|i - j|^\alpha} \tag{3}$$

where the upper bound obeys a power law and that $\epsilon(t) \ll 1$ for $t \lesssim O(S_{dS})$.

We provide a detailed argument in Supplementary Note 3 for certain scale-invariant random MERAs at large bond dimension. Because at each time $t$, the size of $\epsilon$ is suppressed by $\exp(-O(S))$ for this model, the effect of overlaps becomes less visible for universes

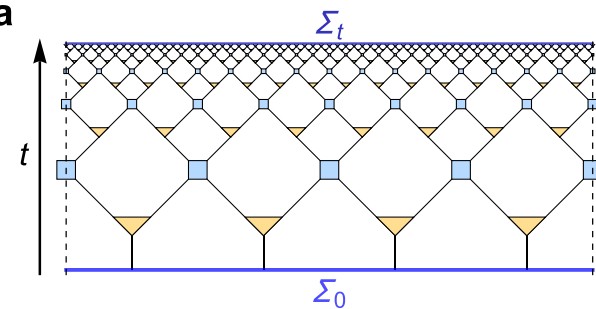

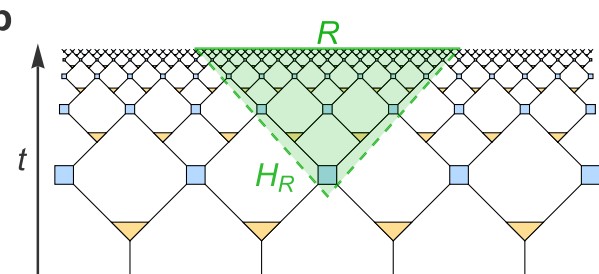

**Fig. 6 | Global and local non-isometric maps in a MERA discretization of de Sitter spacetime. a** In the global proposal, we consider a map $V_{global}$ from the degrees of freedom on a time-slice $\Sigma_t$ at global time $t$ to an initial time-slice $\Sigma_0$. While the Hilbert space dimension on $\Sigma_t$ grows exponentially with $t$, it is constant on $\Sigma_0$. The left and right end of the diagram (dashed lines) are identified. **b** The alternative local proposal involves a map $V_{local}$ from any subregion $R$ (of a time-slice $\Sigma_t$) to the (nearly) light-like horizon $H_R$ of its past domain of dependence, with $H_R$ having exponentially smaller Hilbert space dimension.

with smaller cosmological constants, thereby strengthening the notion of locality in the large $\chi$ limit.

Heuristically, we argue that a similar behavior can be anticipated in the large $\chi$ limit for a less random MERA construction where the tensor network reproduces a CFT ground state. For example, it is expected that for some CFTs, a MERA with large $\chi$ can converge to an exact low energy truncation that preserves the first poly($\chi$) energy eigenstates. Note that this is not guaranteed, however, as the validity of MERA has not been demonstrated for CFTs of large central charges and strong interactions. Recall that the transition $\langle E|O|\Omega\rangle$ induced by a local operator $O$ between some low-energy state $|\Omega\rangle$ and a high energy state $|E\rangle$ with energy eigenvalue above the cutoff is exponentially suppressed by the energy difference between the two states for local Hamiltonians[24,25]. As a larger $\chi$ leads to a higher energy cutoff, the notion of locality (3) becomes sharper as one increases $\chi$ for such critical MERAs also. This is to say that the overlap between spacelike separated observables with respect to sufficiently low energy states vanish in the large $\chi$ limit. We hasten to point out that because the MERA constructions are coarse-grained and have no sub-Hubble features, the overlap within each causal patch (3) does not decay with distance even though $\epsilon$ is small. As a more realistic tensor network model of our universe with sub-Hubble features has not yet been constructed, we will leave this discussion for future work.

At finite sizes, we also verify the distance dependence numerically with examples using a MERA optimized for an Ising CFT (Fig. 7) or using random tensors (Supplementary Note 4). Note that some of the MERA overlaps vanish because of discretization as the past-causal cone of some sites do not overlap with a finite-time cut off. As this computes the commutator norm, it puts an upper bound on the amount of overlap one has for any state.

It is more physically relevant to examine the overlap with respect to a particular state. For example, the commutator is state-dependent

where the amount of overlap weakly correlates with the energy of the state with respect to the Ising CFT Hamiltonian above a certain threshold (Fig. 3b). It is clear that the overlaps with respect to select states are far smaller than the overall norm in Fig. 7. See also Supplementary Fig. 5 for spatial overlap with respect to the ground state.

However, this approximation cannot persist indefinitely as we eventually run out of space to accommodate the overlapping qubits. In units of Hubble time, the approximation is only valid for $T \sim O(S_{dS})$ before $\epsilon \sim O(1)$, a prediction consistent with the scaling found in ref. 22. In Supplementary Note 3, we show that such an estimate is reproduced by a MERA model of the dS static patch. $M$-point functions inside a static patch would hold for a similar time scale up to constant multiplicative factors and logarithmic corrections.

**Proposition 2.** Let $Q^{i_k} \in L(\mathcal{H}_{N_t})$ be observables in the static patch. Then for constant $M$, there exist MERA non-isometric maps $V$ such that $M$-point correlations are well-spoofed with

$$|\langle\psi_p|Q_p^{i_M}\dots Q_p^{i_1}|\psi_p\rangle - \langle\psi|Q^{i_M}\dots Q^{i_1}|\psi\rangle| \sim O(\epsilon) \tag{4}$$

where $\epsilon < O(1)$ up to time $T \lesssim S/2M$.

A similar time scale for $T$ can also be obtained using a JL-based construction in ref. 1 which would permit constant overlap for qubits separated by super-Hubble distances.

More generally, the overlapping qubit construction leads to two notions of dynamics: The first one is from the perspective of the effective description $\mathcal{H}_{N_t}$ where a state with no support in the null space is mapped from an earlier time to the later time by the isometric map $V_t$ constructed by the MERA tensors, whereas the backwards time evolution is co-isometric. In the fundamental Hilbert space $\mathcal{H}_n$, however, we require the evolution to be unitary, with the corresponding Hamiltonian depending on the specifics of dS gravity. Nevertheless, the dynamics described by both should match in the sense that if one pulls back a future state in the effective Hilbert space to the fundamental one via the non-isometric map, it should coincide with the unitarily evolved state in the fundamental Hilbert space.

## Local dS MERA

More recent proposals for a holographic interpretation of dS spacetime have suggested that a holographic theory may be encoded on stretched horizons that are nearly light-like[20,45]. Such an approach readily leads to a second construction for the implementation of a non-isometric map that reduces the apparent number of degrees of freedom on time-slices. We construct a non-isometric map from spatial subregions $R \subset \Sigma_t$ on a dS time-slice $\Sigma_t$ to the fundamental degrees of freedom located at the stretched horizon $H_R$ of its past domain of dependence. Assuming such a map to a fundamental Hilbert space on nearly light-like surfaces is well-defined, we can immediately see from the MERA discretization (Fig. 6b) that it implies an exponential reduction in degrees of freedom. Mirroring the causal structure in the continuum, the wedge $\mathcal{W}_R$ bounded by $\partial\mathcal{W}_R = R \cup H_R$ contains the MERA tensors that are irrelevant to computing expectation values of operators with support only on $R^c$, the complement region to $R$. In particular, if $R$ is chosen to cover half of the global space, its wedge $\mathcal{W}_R$ corresponds to the "exterior region" between two initial static patches in $1+1$ dimensions. $H_R$ then overlaps with the horizons of both patches. Interestingly, the entanglement structure is altered by the compression map $V_{local}$, such that it converts a non-flat continuous spectrum for a state in the exterior region to one of nearly maximally entangled pairs along the stretched horizon. As overlaps are analogous to perturbative gravitational effects, in the infinite dimensional limit this is qualitatively similar to[46], where it was found that gravitational effects convert a type III von Neumann algebra type II$_1$, even though the conversion occurred on different spacetime regions. We further comment on this in Supplementary Note 5.

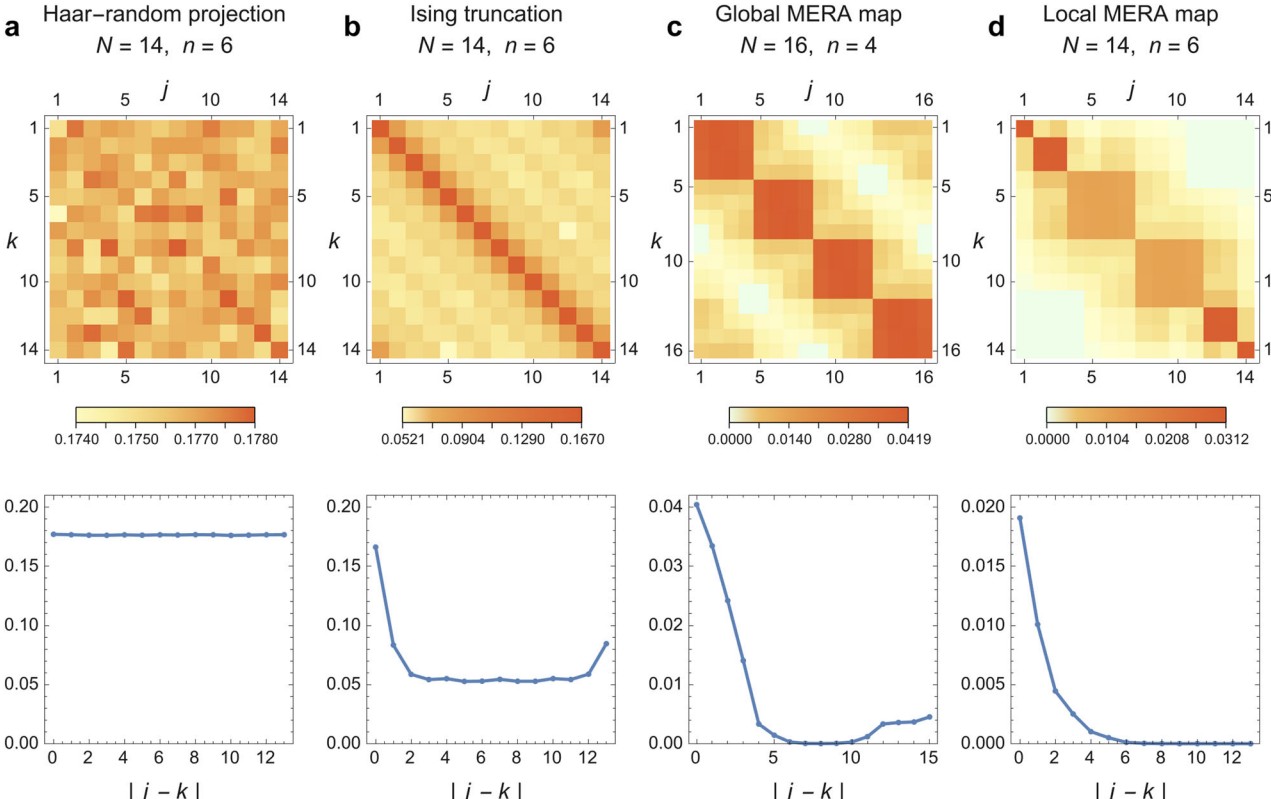

**Fig. 7 | Model-dependent non-commutativity of Pauli operators.** All plots show the normalized trace norm of the commutator $\| [X_p^{(j)}, Z_p^{(k)}] \|_1 / \| X_p^{(j)} \|_1 \| Z_p^{(k)} \|_1$ between projected Pauli operators $X_p^{(j)} = VX^{(j)}V^\dagger$ and $Z_p^{(k)} = VZ^{(k)}V^\dagger$. Here $V : \mathcal{H}_N \to \mathcal{H}_n$ is a model-dependent non-isometric map from an $N$-dimensional to a smaller $n$-dimensional Hilbert space. The considered model for $V$ are (**a**) an Akers-Pennington type Haar-random projection (averaged over 20 samples), (**b**) a Hilbert space truncation to the low-energy critical Ising model, (**c**) a local MERA map, and (**d**) a global MERA map. The MERA maps, shown in Fig. 6, use tensors optimized with respect to the critical Ising model ground state. The dependence of the average commutator norm on distance $|j - k|$ is plotted in the second row of each plot, showing a decay for all but the Haar-random projection. The Ising truncation leads to a plateau at large distances, but allows for a stronger state-dependent decay as shown in Supplementary Fig. 5.

The behavior of commutators is quite similar between the global and local map: As we find in Fig. 7, both show a decay of the trace norm of the commutator $\| [X_p^{(j)}, Z_p^{(k)}] \|_1$ of two projected $X$ and $Z$ operators with distance $|j - k|$, such that commutativity is approximately restored at large distances. By construction, the global map exhibits periodic boundary conditions, while the local map does not. Using Haar-random disentanglers/isometries yields qualitatively similar results discussed in Supplementary Note 4. The map $V_{\text{local}}$ implemented by the local MERA picture, however, relates two very different types of theories and their states: Unlike in the global MERA, whose $V_{\text{global}}$ effectively describes a locality-preserving RG map between critical theories at different length and energy scales, $V_{\text{local}}$ maps local operators on $H_R$ to both local and highly nonlocal operators on $R$, depending at which time $t$ the operator is inserted on the horizon. The effect of applying the projector $P = V^\dagger V$ on a critical theory for $V = V_{\text{global}}$ results in a projection to its low-energy subspace. We can confirm numerically in Supplementary Note 4 that only the first $\sim 2^n$ eigenstates are preserved with meaningful norm. However, choosing a $V = V_{\text{local}}$ leads to a much slower loss in fidelity as one moves to higher-energy states, thus approximately preserving a large part of the low-energy subspace of the theory (in our example, the critical Ising model). In a quantum gravity interpretation of this model, the effective Hilbert space is thus softly truncated at high energies, as is generally expected due to black hole contributions.

Interestingly, a similar construction using non-isometric maps in refs. 10,13 produced a similar truncation for the Hilbert space of the black hole interior, but with regard to states of high *complexity* rather than energy. Using Haar-random projections, it was found that the complexity needed for operations to reach the null space (or kernel) is exponential in the entropy of the black hole. This permits the effective theory to remain approximately valid for all subexponential processes. Similarly, we can easily identify processes that interpolate between physical states and null states in MERA. This can be informative as it tells us how complex an operation would be needed before our effective theory predictions break down. In both the global and local models, the null states describe excitations where the isometry is written as a unitary with partial projection onto an ancilla state orthogonal to $|0\rangle$. To produce a null state in the MERA circuit, one thus needs to "undo" some gates in the future lightcone of any isometry, change its ancillary projection, and apply the gates in reverse, a process visualized in Fig. 8. Thus if the complexity of a single tensor is $C$, then the complexity to reach any null state is $\le kC$ where $k$ is upper bounded by the spacetime volume (in Hubble units) of the future lightcone emanating from the altered ancilla and terminating at the late time cutoff. The complexity $C \sim \exp(S_{dS})$ if the tensors themselves are Haar-random, but can be $\text{poly}(S_{dS})$ if they are generated by some local theory or ordinary time evolution in the EFT.

## Discussion

Our work proposes that there exists a deep relationship between the Hilbert space dimension verification problem in quantum information and the holographic principle in quantum gravity. With a construction of overlapping qubits based on a local tensor network, we created approximate qubits that have distance-dependent overlaps as opposed to ones that are pair-wise constant[1,13], a relevant property for physical settings. The specific application of non-isometric maps to dS/

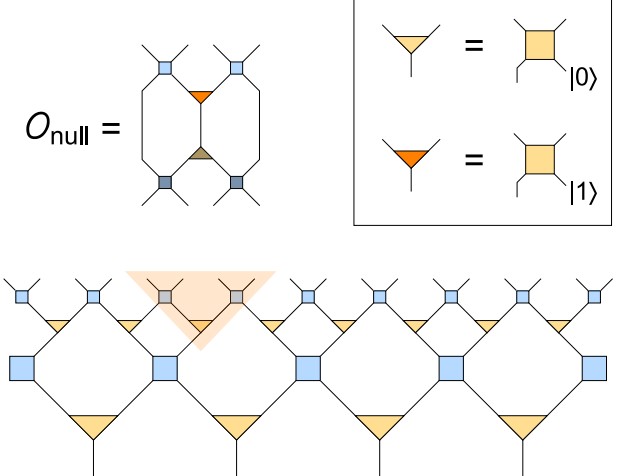

**Fig. 8 | Accessing the null space of the tensor network map with an operator.** This operator $\mathcal{O}_{null}$ is constructed by undoing disentanglers and isometries in a wedge (shaded region) by applying their conjugates (dark tensors as in Fig. 5), and then applying a modified isometry orthogonal to the original (orange tensor). As shown in the legend, this modified isometry follows from the representation of the original isometry as a unitary map with projection onto an ancilla state $|0\rangle$, which is flipped to $|1\rangle$. Operators $\mathcal{O}_{null}$ can be constructed relative to any isometry in the tensor network, with lower-lying isometries leading to higher operator circuit complexity.

MERA relates non-local dynamical processes over the fundamental degrees of freedom to quasi-local processes over a larger set of apparent degrees of freedom, reminiscent of ref. 21. This provides an explicit mapping that reconstructs exterior geometry from horizon degrees of freedom in the local MERA model, delivers insight on dS quantum gravity, and connects with discussions in eternal inflation and cosmology[22]. The non-isometricity of the Hilbert space map $V$ also indicates that the theory is unitarity-violating on at least one side of the duality, albeit weakly in the well-spoofed limit. Though not a required feature for producing overlapping qubits, our specific examples of $V$ are *co-isometric*, i.e., with $V^\dagger$ forming an isometry. While natural for describing global dS evolution[26], it is unclear if this property is always necessary in Hilbert space truncations due to quantum gravity.

Although our explicit construction of overlapping qubits differs from that in ref. 1, which can be unmasked with linearly many operations in $N$, it is worth commenting on the complexity of such a protocol in a larger gravitational context. For instance, a naive protocol in the actual universe that involves volumetrically many apparent EFT degrees of freedom has the potential to create significant gravitational back-reaction by forming black holes. Therefore it is entirely reasonable from gravitational considerations that operations polynomial in $N$ can be used to verify the dimension of the fundamental Hilbert space. However, the precise scaling is likely relevant for a careful consistency check, which would constitute an interesting future direction.

The idea of spoofing opens up new directions of reformulating quantum field theories in the overlapping qubit language by relaxing the commutation relations for spacelike separated operators in canonical quantization and exploring their phenomenological implications in particle physics and cosmology. When applied to quantum gravity, it remains open whether the overlaps introduced by gravitational dressing can resolve the degree of freedom counting problem in holography beyond AdS/CFT.

From a complexity theoretic point of view, actual experiments and physical processes are problems that can be solved or verified with polynomial (time or space) complexity (e.g., NP, QMA). However, for problems not belonging to such classes, many distinct processes may appear indistinguishable when a system is probed with limited computational resources[10,47,48]. The overlapping qubit approach adds to the

list yet another instance, where the resource-intensive task is the verification of the exact commutativity between space-like separated operators. It thus contributes to an emerging program of understanding quantum gravity from an accuracy- and resource-limited perspective.

## Data availability
The authors declare that the data supporting the findings of this study are available within the paper. The numerical data shown in figures is available from the corresponding author on request.

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

## Acknowledgements

We thank Scott Aaronson, Chris Akers, Vijay Balasubramanian, Adam Brown, Oliver Friedrich, Elliott Gesteau, Philipp Höhn, John Preskill, Suvrat Raju, Leonard Susskind, Brian Swingle, Yu Tong, Thomas Vidick, Jinzhao Wang, and Zhenbin Yang for interesting comments and discussions. A.J. was supported by the Simons Collaboration on It from Qubit, the US Department of Energy (DE-SC0018407), and the Einstein Research Unit "Perspectives of a quantum digital transformation". C.C. acknowledges the support by the U.S. Department of Defense and NIST through the Hartree Postdoctoral Fellowship at QuICS, the Air Force Office of Scientific Research (FA9550-19-1-0360), and the National Science Foundation (PHY-1733907). Z.Z. was supported by the National Research, Development, and Innovation Office through the Quantum Information National Laboratory of Hungary (Grant No. 2022-2.1.1-NL-2022-00004) and Grant No. FK135220.

## Author contributions

C.C., W.C., A.J., and Z.Z. jointly participated in the conception and discussion of the work. C.C. and A.J. are primarily responsible for the technical content and writing of the paper.

## Funding

## Competing interests

The authors declare no competing interests.
