## [Transparent Peer Review file · Nature Communications]

Overlapping qubits from non-isometric maps and de Sitter tensor networks

Corresponding Author: Dr Alexander Jahn

Version 0:

Reviewer comments:

Reviewer #1

(Remarks to the Author)
Dear Editor,

The present article aims to study non-locality and isometric time evolution in models of de Sitter quantum gravity. (Small amounts of) Non-locality is expected to be a general feature in consistent theories of quantum gravity, although the precise details are still a matter of vigorous debate, while isometric bulk-to-boundary codes have been en vogue as a way of thinking about black hole complementarity and the bulk-to-boundary correspondence. The authors study both of these properties in the context of MERA constructions, which they claim approximate physics in two-dimensional de Sitter universes, and this in turn is why they claim their results describe features of quantum gravity in that setting. Moreover there is the very important matter of the Hilbert space dimension, expected to be finite for dS gravity (more on that below), which the authors attempt to address in the context of their MERA constructions.

This is, quite frankly, a very difficult paper to evaluate. First, the positives. Within their ansatz, the authors (as far as I can tell) correctly evaluate some interesting probes of non-locality which make some intuitive sense for de Sitter physics. They also try to simultaneously describe global and local aspects of dS quantum gravity, e.g. the Hilbert space dimensions, for which the gravity results they are aiming to match are presently unknown. On the one hand, this is a real feather in the authors' cap, since they are attempting to address questions which we cannot presently address in a completely tractable and principled way. On the other this makes it difficult to know if the work is really about gravity or if it is just about tensor networks and no more.

That brings me to what I consider to be some negative aspects of the paper, having to do with the connections between their ansatz and de Sitter gravity. (Here I will have more to say than the positives not because my opinion is more negative than positive — it's not! — but rather in the interest of constructive criticism.) After all, recall that the authors study tensor networks which have no a priori reason to accurately model gravity; the basic idea of their approach is to very carefully treat MERA schemes, and then make physical arguments that the results have something to do with de Sitter. But there I simply am concerned with those arguments.

The most basic issue I have is that the proposed finiteness of the dS gravity Hilbert space appears in two different settings in the gravity literature, neither of which as I understand it matches what the authors find. The first is the idea, suggested by the entropy of the causal horizon of the static patch and the minus sign of the gravitational first law therein (and more recent constructions of algebras of observable for semi classical gravity), that the Hilbert space of the static patch is finite-dimensional with dimension $\exp(S_{\text{dS}})$. The second, going back to Witten's thoughts about a prospective dual to de Sitter gravity, is that the S-matrix of dS gravity (including an asymptotic past region) has a co-kernel of dimension $\exp(S_{\text{dS}})$. To the extent that the authors try to physically motivate a connection between MERA and gravity models in terms of tracking the Hilbert space dimensions, I would think that such a connection should be taken seriously only if it matches that phenomenology.

I have other issues that lead me to question the whole enterprise. The idea that the dimension of the bulk Hilbert space depends on time seems wrong to me; that is true in a lattice field theory on an expanding universe, but quantum gravity crucially does not admit such a lattice regularization. At an even more basic level, the authors propose to describe global de Sitter space, but the past half is missing. There are also these statements about nonzero locality even in perturbative gravity,

which are not quite true. (The Papadodimas/Raju proposal for example applies in very excited states, black holes; about the vacuum non-locality isn't expected to appear until the string scale, and gravitational dressing certainly does not generate such non-locality either, since the relevant Wilson lines can always be chosen to not cross.) Lastly, I have a problem with the whole enterprise of using isometric/non-isometric codes as a proxy for problems in quantum gravity; I am content to regard it as an interesting point of research into problems in gravity where no principled and tractable approach currently exists, as long as the qualifiers are given that it is a completely unsystematic and inspired ad hoc model for bulk physics.

However my complaints are to some extent not with the authors' work but rather with the entire subregion in which the work lives. For that reason you should not interpret my criticisms as harsh judgment, but rather skepticism about the larger program, of which (independent of my criticisms) the present work is a much-better-than-average representative.

In light of all of these comments, unfortunately I am not sure what to recommend when it comes to publication. I have already presented my complaints above. The argument in favor of publication, provided that the work can be more carefully qualified than it is now, is that the problems the authors are trying to address are important, long-standing, and seemingly impenetrable with gravity methods, justifying the use of an ad hoc model. I think I could buy that argument, provided that again the work is suitably qualified.

What I would really like however goes far beyond the work in its present form. Importantly I think that since the authors spend so much effort to keep track of the dimension of various Hilbert spaces, that they should be able to see $S_{\{dS\}}$ in either the static patch entropy or in the global dS S-matrix, as I complained above.

Reviewer #2

(Remarks to the Author)

This is a referee report for "Overlapping qubits from non-isometric maps and de Sitter tensor networks" by Cao, Chemissany, Jahn, and Zimborás. The manuscript makes connections between recent work in quantum information concerning the "spoofing" of a larger-dimensional Hilbert space of qubits by means of a smaller actual Hilbert space, and the appearance of non-isometric error correcting codes in quantum gravity as a means of implementing holography. This provides another approach to these non-isometric maps beyond the Haar-random unitaries of Akers et al. The authors furthermore provide a realization of these ideas in a MERA network model of de Sitter space.

This paper is very relevant and potentially has large application to the program of understanding quantum gravity and holography through tools adopted from quantum information theory. Non-isometric maps have emerged as vital for this enterprise and the presentation of a new set of tools for realizing them has obvious relevance. I would be interested to see similar work using these tools for the black hole interior. I recommend the paper's publication in Nature Communications.

Reviewer #3

(Remarks to the Author)

In the manuscript, the authors propose a novel framework for approximate, overlapping qubits as a possible solution to the problem of degrees of freedom counting occurring whenever a holographic relation is postulated between an effective (field) theory and a fundamental quantum gravity description. In a very generic setting, the work shows that it is indeed possible to fit $N > n$ approximate overlapping qubits into the fundamental Hilbert space, over which a quasi-local and approximately unitary effective theory can be defined, through a non-isometric map within some well-defined (energy) regime.

In particular, the overlap is quantified in the work by non-commutativity between space-like separated operators. The authors provide examples based on a local tensor network, where approximate qubits with distance-dependent overlaps are created. Concrete realisation of distance-dependent overlap in MERA-like states is the main noteworthy result of the work.

Building on the dS/MERA correspondence, the work stresses the importance of approximate orthogonality in the context of quantum gravity and cosmology. The authors claim that the overlap between approximate qubits may be generically identified with the weak non-locality expected in gauge theory, such as the non-local effects (micro-causality breaking) introduced by the gravitational dressing. Interestingly, the overlapping qubits resulting from a non-isometric truncation of EFT Hilbert spaces produce analogous effects with those of quantum gravity.

However, despite being supported by similar ideas in recent literature, besides plausible reasoning, the work does not reinforce the claim via a concrete analysis. That may limit the significance of the proposed idea to the field.

For instance, it is unclear whether one should understand the Hilbert space of the effective theory as a physical or a kinematic Hilbert space. In the global dS MERA example, if the notion of "evolving" Hilbert spaces had to do just with the varying number of degrees of freedom of the underlying discrete network, then the kinematic Hilbert spaces on the (time-evolved) refined graph would be isometrically isomorphic to the fundamental one even if $n \neq N$. The actual change of dimension seems to be related to gravitational dynamics and somehow correlated with the variation of the entanglement scale. Is this effective dynamics one should intend as being approximated by a sequence of non-isometric maps? A few more details could help understanding.

Also, the relevant comment on the null-states production at the end of the local dS-MERA example could be made more explicit with a direct computation.

Despite the comments, the work is interesting for a broad audience; the content is timely in quantum gravity. There are no flaws in the derivations and the data analysis. The methodology is sound. Overall, the work meets the expected standards in the field.

Response to Reviewers

Reviewer #1 (Remarks to the Author):

Dear Editor,

The present article aims to study non-locality and isometric time evolution in models of de Sitter quantum gravity. (Small amounts of) Non-locality is expected to be a general feature in consistent theories of quantum gravity, although the precise details are still a matter of vigorous debate, while isometric bulk-to-boundary codes have been en vogue as a way of thinking about black hole complementarity and the bulk-to-boundary correspondence. The authors study both of these properties in the context of MERA constructions, which they claim approximate physics in two-dimensional de Sitter universes, and this in turn is why they claim their results describe features of quantum gravity in that setting. Moreover there is the very important matter of the Hilbert space dimension, expected to be finite for dS gravity (more on that below), which the authors attempt to address in the context of their MERA constructions.

We fully agree with the Reviewer.

This is, quite frankly, a very difficult paper to evaluate. First, the positives. Within their ansatze, the authors (as far as I can tell) correctly evaluate some interesting probes of non-locality which make some intuitive sense for de Sitter physics. They also try to simultaneously describe global and local aspects of dS quantum gravity, e.g. the Hilbert space dimensions, for which the gravity results they are aiming to match are presently unknown. On the one hand, this is a real feather in the authors' cap, since they are attempting to address questions which we cannot presently address in a completely tractable and principled way. On the other this makes it difficult to know if the work is really about gravity or if it is just about tensor networks and no more.

We thank the Reviewer for their positive comments. We would like to stress that our approach is agnostic about full de Sitter quantum gravity, only assuming the constraint that the effective semiclassical Hilbert space must remain at constant size and exploring the consequences on operator locality in that effective description.

That brings me to what I consider to be some negative aspects of the paper, having to do with the connections between their ansatze and de Sitter gravity. (Here I will have more to say than the positives not because my opinion is more negative than positive — it's not! — but rather in the interest of constructive criticism.) After all, recall that the authors study tensor networks which have no a priori reason to accurately model gravity; the

basic idea of their approach is to very carefully treat MERA schemes, and then make physical arguments that the results have something to do with de Sitter. But there I simply am concerned with those arguments.

Our use of MERA to describe the semi-classical degrees of freedom of de Sitter gravity builds on previous work by Refs. [19,35,36,38]. Indeed, the idea that de Sitter time evolution follows an isometric map between effective Hilbert spaces has been more generally proposed and explored in Refs. [26-28]. We have added the following clarifying sentence in Sec. III:

“The MERA thus parametrizes the isometric map that appears between effective degrees of freedom on these time-slices [26-28] while incorporating causal constraints:” (page 5)

However, we completely agree that full quantum gravity is likely outside of the scope of tensor network methods, and have added in Sec. III:

“We also hasten to point out that the dS/MERA construction serves as a coarse-grained mapping for how the quasi-local EFT degrees of freedom on the (super) Hubble scale that increase in time emerge from a smaller “quantum gravity” Hilbert space where the fundamental theory can be highly non-local. However, it does not constitute a theory of de Sitter quantum gravity on the fundamental Hilbert space.” (page 5-6)

The most basic issue I have is that the proposed finiteness of the dS gravity Hilbert space appears in two different settings in the gravity literature, neither of which as I understand it matches what the authors find. The first is the idea, suggested by the entropy of the causal horizon of the static patch and the minus sign of the gravitational first law therein (and more recent constructions of algebras of observable for semi classical gravity), that the Hilbert space of the static patch is finite-dimensional with dimension $\exp(S_{dS})$. The second, going back to Witten’s thoughts about a prospective dual to de Sitter gravity, is that the S-matrix of dS gravity (including an asymptotic past region) has a co-kernel of dimension $\exp(S_{dS})$. To the extent that the authors try to physically motivate a connection between MERA and gravity models in terms of tracking the Hilbert space dimensions, I would think that such a connection should be taken seriously only if it matches that phenomenology.

We agree with the referee that a more detailed model and a more systematic approach that also recovers the existing phenomenology of dS quantum gravity would be highly desirable. We also agree that although many phenomenological constraints on our dS/MERA model have been inspired by existing proposals in dS quantum gravity, some of its features are not identical to the proposals commonly found in dS gravity literature.

Regarding the finiteness of de Sitter entropy, the two observations mentioned by the Reviewer appear as follows in our construction: 1. The discrete analogue of the static patch in our model is a vertical strip of tensors, alternating between a disentangler and two isometry tensors. Its

entropy is precisely S_{dS} which is set by the bond dimension of the tensor contractions. We have added these sentences to the manuscript:

“This is the discrete analogue of the static patch in the tensor network model where it corresponds to a vertical strip of tensors, alternating between a disentangler and two isometry tensors.” (page 5)

“Therefore, the entropy of each static patch is precisely $O(S_{dS})$, set by the bond dimension of the tensor contractions.” (page 5)

2. A restriction to the entropy of the co-kernel of the de Sitter S-matrix (from an asymptotic past to an asymptotic future) is the central feature of our “global” overlapping qubit proposal, in which we restrict the EFT Hilbert space to that of the de Sitter throat region at $t=0$. From this, the entropy of a single static patch is indeed S_{dS} , consistent with the proposed value of static patch entropy in the first proposal above. To clarify, we have added the following to the manuscript:

“Such an entropic restriction is also consistent with previous studies of de Sitter quantum gravity, the co-kernel of whose time evolution is restricted to states at the throat region at $t = 0$ [38, 40]. For a single static patch, this leads to the expected entropy S_{dS} .” (page 6)

I have other issues that lead me to question the whole enterprise. The idea that the dimension of the bulk Hilbert space depends on time seems wrong to me; that is true in a lattice field theory on an expanding universe, but quantum gravity crucially does not admit such a lattice regularization.

We completely agree with the referee that *“The idea that the dimension of the bulk Hilbert space depends on time [...] is true in a lattice field theory on an expanding universe, but quantum gravity crucially does not admit such a lattice regularization”*. Indeed the bulk Hilbert space corresponds to the latticization of the effective field theory which does grow in time in a de Sitter background. However, the fundamental quantum gravity Hilbert space remains fixed in size in our model [c.f. 1st paragraph, section “Global dS MERA”, page 6]. One purpose of our work is precisely to use MERA to reconcile these two pictures where one can think of MERA as the map that tells one how to compress the bulk EFT degrees of freedom into the fundamental QG Hilbert space (or conversely how the EFT emerges from the fundamental degrees of freedom). The growing Hilbert space on the latticized network is describing semi-classical dS, not quantum gravity.

At an even more basic level, the authors propose to describe global de Sitter space, but the past half is missing.

A time-symmetric version of the dS/MERA exists but we only focus on the top half of the tensor network as that's the part that's relevant for our work. A related explicit construction is given by [38], which considers an encoding map from the asymptotic past to the asymptotic future. The code space restriction of this Hilbert space is similar to our global dS proposal. Our newly added comment on page 6 (mentioned above) now addresses features of this construction.

There are also these statements about nonzero locality even in perturbative gravity, which are not quite true. (The Papadodimas/Raju proposal for example applies in very excited states, black holes; about the vacuum non-locality isn't expected to appear until the string scale, and gravitational dressing certainly does not generate such non-locality either, since the relevant Wilson lines can always be chosen to not cross.)

We would like to clarify that the statement we make about non-zero locality is a state-dependent statement where in certain constructions, such as the examples shown (e.g. Figure 3) the overlap indeed vanishes for sufficiently low energy states. We agree with the referee that the comment about non-locality in perturbative gravity (for some choice of dressing) is misleading and actually unnecessary for our purposes. To remove this ambiguity, we have removed a sentence about gravitational dressing from the introduction (page 2) and instead modified and extended the following clarification in Sec. 2:

"If Q, Q' are gauge invariant operators [18] that are spacelike separated, then the small but non-vanishing commutator of $[Q_p, Q_{p'}]|\psi\rangle$ is consistent with the them being weakly non-local by turning on gravity. Formally, $T^{\wedge}[Q, Q']_p$ plays the role of the gravitational correction, where it is generally a non-local operator straddling the local operators Q, Q' . Note that the gravitational correction can admit both perturbative and non-perturbative contributions. However, the perturbative effect may depend on the choice of the gravitational dressing [18]."
(page 3)

Lastly, I have a problem with the whole enterprise of using isometric/non-isometric codes as a proxy for problems in quantum gravity; I am content to regard it as an interesting point of research into problems in gravity where no principled and tractable approach currently exists, as long as the qualifiers are given that it is a completely unsystematic and inspired ad hoc model for bulk physics. However my complaints are to some extent not with the authors' work but rather with the entire subregion in which the work lives. For that reason you should not interpret my criticisms as harsh judgment, but rather skepticism about the larger program, of which (independent of my criticisms) the present work is a much-better-than-average representative.

In light of all of these comments, unfortunately I am not sure what to recommend when it comes to publication. I have already presented my complaints above. The argument in favor of publication, provided that the work can be more carefully qualified than it is now, is that the problems the authors are trying to address are important, long-standing,

and seemingly impenetrable with gravity methods, justifying the use of an ad hoc model. I think I could buy that argument, provided that again the work is suitably qualified.

We agree that the use of quantum information methods is a promising approach towards quantum gravity that offers insights to questions that are hard to tackle through traditional gravitational approaches, even though many current proposals rely heavily on ad hoc models. However, we believe that our tensor network setup is well-justified in the semi-classical setting we consider, where it appears from a systematic discretization of EFT degrees of freedom. The ad hoc part of our work is merely the assumption that the (unknown) Hilbert space of de Sitter quantum gravity is finite, implying a finite subspace of semi-classical states as well. We then show that such a Hilbert space truncation can almost exactly preserve operator locality, though the exact form of this truncation map (e.g. the choice of tensors in the tensor network) will depend on many additional physical assumptions that we do not explicitly consider. As we find in our work, this main result holds regardless of such specific choices, and holds even for Haar-random tensors. To further qualify our work and the setting in which it applies, we have now added a number of clarifications (see comments above). Regarding the scope of the dS/MERA approach, we also added the following disclaimers to the Introduction and the section “Review of de Sitter MERA”:

“Tensor networks with suitable geometries are a natural discretization of the maps between EFT Hilbert spaces on different time-slices, though it is at present unclear if they can also describe features of full, nonperturbative quantum gravity.” (page 2)

“We qualify that the dS/MERA construction only serves as a coarse-grained mapping for how the quasi-local EFT degrees of freedom on the (super) Hubble scale that increase in time emerge from a smaller “quantum gravity” Hilbert space where the fundamental theory can be highly non-local. It does not constitute a theory of de Sitter quantum gravity.” (page 6)

What I would really like however goes far beyond the work in its present form. Importantly I think that since the authors spend so much effort to keep track of the dimension of various Hilbert spaces, that they should be able to see $S_{\{dS\}}$ in either the static patch entropy or in the global dS S-matrix, as I complained above.

As written above, the de Sitter entropy indeed appears both in the discretized static patch as well as in the global proposal, relating to previous (quantum) gravity proposals of de Sitter spacetime. We also note that the S-matrix formalism cannot be straightforwardly applied to tensor networks as there is no natural notion of asymptotically free states. The precise connections of our tensor network formalism to quantum gravity proposals for de Sitter will be an interesting question to explore in future work.

Reviewer #2 (Remarks to the Author):

This is a referee report for "Overlapping qubits from non-isometric maps and de Sitter tensor networks" by Cao, Chemissany, Jahn, and Zimborás. The manuscript makes connections between recent work in quantum information concerning the "spoofing" of a larger-dimensional Hilbert space of qubits by means of a smaller actual Hilbert space, and the appearance of non-isometric error correcting codes in quantum gravity as a means of implementing holography. This provides another approach to these non-isometric maps beyond the Haar-random unitaries of Akers et al. The authors furthermore provide a realization of these ideas in a MERA network model of de Sitter space.

We agree with the Reviewer's summary of our work.

This paper is very relevant and potentially has large application to the program of understanding quantum gravity and holography through tools adopted from quantum information theory. Non-isometric maps have emerged as vital for this enterprise and the presentation of a new set of tools for realizing them has obvious relevance. I would be interested to see similar work using these tools for the black hole interior. I recommend the paper's publication in Nature Communications.

We thank the Reviewer for their positive appraisal and recommendation for publication.

Reviewer #3 (Remarks to the Author):

In the manuscript, the authors propose a novel framework for approximate, overlapping qubits as a possible solution to the problem of degrees of freedom counting occurring whenever a holographic relation is postulated between an effective (field) theory and a fundamental quantum gravity description. In a very generic setting, the work shows that it is indeed possible to fit $N > n$ approximate overlapping qubits into the fundamental Hilbert space, over which a quasi-local and approximately unitary effective theory can be defined, through a non-isometric map within some well-defined (energy) regime.

In particular, the overlap is quantified in the work by non-commutativity between space-like separated operators. The authors provide examples based on a local tensor network, where approximate qubits with distance-dependent overlaps are created. Concrete realisation of distance-dependent overlap in MERA-like states is the main noteworthy result of the work.

Building on the dS/MERA correspondence, the work stresses the importance of approximate orthogonality in the context of quantum gravity and cosmology. The authors claim that the overlap between approximate qubits may be generically identified with the weak non-locality expected in gauge theory, such as the non-local effects (micro-causality breaking) introduced by the gravitational dressing. Interestingly, the overlapping qubits resulting from a non-isometric truncation of EFT Hilbert spaces produce analogous effects with those of quantum gravity.

We agree with the Reviewer's summary of our work.

However, despite being supported by similar ideas in recent literature, besides plausible reasoning, the work does not reinforce the claim via a concrete analysis. That may limit the significance of the proposed idea to the field. For instance, it is unclear whether one should understand the Hilbert space of the effective theory as a physical or a kinematic Hilbert space.

We thank the Reviewer for this constructive comment and the stimulating question. The key objective of this paper is to demonstrate that locality can be approximately preserved under non-isometric mappings and a more physically relevant distance-dependent overlap is achievable. For both of these claims, we backed up both with small scale numerics and analytic MERA-based tensor network analysis for large system sizes. As both numerical and analytic results indicate that this is clearly possible and true, we would like to clarify that our claim has indeed already been backed up by a concrete analysis.

As a corollary of that objective, we discuss the implication of such compression maps in the context of dS/MERA correspondence, which we agree is far more interesting for quantum

gravity, but also less concrete at the current stage. We also agree that a more precise correspondence with quantum reference frames (QRF) is highly relevant for the community, but it would be beyond the scope of our current work. However, as tensor network models in general still lack the level of rigor one enjoys in the rest of high energy theory, a proper identification of gauge constraints in a dS tensor network is still very much an open question where such an analysis, while highly desirable, would merit its own paper.

Nevertheless, we can try to make a few clarifying remarks in this regard. It is plausible that the physical Hilbert space of a QRF formulation could be identified with the fundamental fully gauge invariant quantum gravity Hilbert space. However, it is unclear whether a straightforward connection exists between kinematic or effective Hilbert spaces in this MERA picture. To discuss kinematic Hilbert space, we need to first specify a theory: a quantum field theory on an expanding spacetime with possible coupling to perturbative quantum gravity or a fully UV complete theory of de Sitter quantum gravity. What should be considered as a gauge constraint will likely be different. The effective Hilbert space in our discussion simply refers to the Hilbert space in which a low energy effective theory lives. For concreteness, one may think of it as the Hilbert space of a quantum field theory living on a dS background even though the correspondence is only analogous as we talk about tensor networks. The EFT as defined in this Hilbert space is a conventional one that does not violate microcausality for any state. The compressed observable supported on the fundamental Hilbert space does, however, at least for some states. In this sense, it is tempting to think that the Hilbert space of the low energy EFT has a non-trivial intersection with the kinematic Hilbert space of a local theory before gauge constraints (from perturbative quantum gravity) have been imposed. However, in the absence of a complete theory of quantum gravity that recovers de Sitter as a low energy effective description, we cannot definitively say whether or what constraints one has to impose in QRF such that the physical Hilbert space obtained in this manner can be identified with the fundamental Hilbert space in our construction.

In the global ds MERA example, if the notion of “evolving” Hilbert spaces had to do just with the varying number of degrees of freedom of the underlying discrete network, then the kinematic Hilbert spaces on the (time-evolved) refined graph would be isometrically isomorphic to the fundamental one even if $n \neq N$.

A form of this is indeed manifested in our tensor network. The MERA network is isometric as a map from the past to the future and co-isometric from the future to the past, meaning that the state living in late time that is not in the null space can be identified isometrically with a state in the fundamental/physical Hilbert space. In this sense, the orthogonal complement of the null space (co-kernel) is isomorphic to the fundamental one.

The actual change of dimension seems to be related to gravitational dynamics and somehow correlated with the variation of the entanglement scale. Is this effective dynamics one should intend as being approximated by a sequence of non-isometric maps? A few more details could help understanding.

This is indeed an interesting aspect of the overlapping qubits construction, as there are two ways to understand dynamics. We explain these in a new paragraph in the section “Global dS MERA”:
“More generally, the overlapping qubit construction leads to two notions of dynamics: The first one is from the perspective of the effective description H_{N_t} where a state with no support in the null space is mapped from an earlier time to the later time by the isometric map V_t constructed by the MERA tensors, whereas the backwards time evolution is co-isometric. In the fundamental Hilbert space H_n , however, we require the evolution to be unitary, with the corresponding Hamiltonian depending on the specifics of de Sitter gravity. Nevertheless, the dynamics described by both should match in the sense that if one pulls back a future state in the effective Hilbert space to the fundamental one via the non-isometric map, it should coincide with the unitarily evolved state in the fundamental Hilbert space.” (page 7)

Also, the relevant comment on the null-states production at the end of the local dS-MERA example could be made more explicit with a direct computation.

To show the general construction of a null-state accessing operator, we have added a new figure (Fig. 8) to visualize how such operators can be composed from the MERA tensors.

Despite the comments, the work is interesting for a broad audience; the content is timely in quantum gravity. There are no flaws in the derivations and the data analysis. The methodology is sound. Overall, the work meets the expected standards in the field.

We thank the Reviewer for their positive comments.